# Effect of pH Change on the Microalgae-Based Biogas Upgrading Process

Leslie Meier [1,2], Carlos Vilchez [3] , María Cuaresma [3] , Álvaro Torres-Aravena [4,*] and David Jeison [4]

1. Department of Chemical Engineering, Universidad de La Frontera, Av. Francisco Salazar, Temuco 01145, Chile
2. Scientific and Technological Bioresource Nucleus, Universidad de La Frontera, Av. Francisco Salazar, Temuco 01145, Chile
3. Algal Biotechnology Group, CIDERTA and Faculty of Experimental Sciences, University of Huelva, Huelva Business Park, 21007 Huelva, Spain
4. Escuela de Ingeniería Bioquímica, Facultad de Ingeniería, Pontificia Universidad Católica de Valparaíso, Av. Brasil 2085, Valparaíso 2950, Chile
* Correspondence: alvaro.torres@pucv.cl

**Abstract:** An alternative way to remove $CO_2$ from biogas is the use of photosynthetic microorganisms, such as microalgae. This can be achieved by the operation of an open photobioreactor, connected with a mass transfer column, such as a counterflow column. This technology provides up-graded biogas with high quality. The microalgal uptake of $CO_2$ from the biogas in counterflow columns generates pH changes in microalgae culture. To clarify the potential effect of these dynamic pH conditions in the culture, the effect of pH change on the photosynthetic activity and PSII quantum yield was studied for microalgae *Chlorella sorokiniana*. Thus, assays were carried out, where the pH drop reported in the counterflow columns was replicated in batch microalgae culture through HCl addition and $CO_2$ injection, moving the culture pH from 7.0 to 5.0 and from 7.0 to 5.8, respectively. Moreover, the effect of light/darkness on photosynthetic activity was tested when the pH decreased. The results obtained in this research showed that the photosynthetic activity decreased for the light conditions when the pH was shifted by HCl addition and $CO_2$ injection. Despite this, the value of the PSII quantum yield remained at 0.6–0.7, which means that the microalgae culture did not suffer a negative effect on the photosynthetic system of cells because a high value of PSII efficiency remained. In the same way, the results indicated that when the pH change was corrected, the photosynthetic activity recovered. Moreover, the apparent affinity constant for dissolved inorganic carbon (KDIC) was 0.9 μM at pH 5 and 112.0 μM at pH 7, which suggests that the preferred carbon source for *C.sorokniana* is $CO_2$. Finally, all the results obtained indicated that the pH drop in the counter-flow column for biogas upgrading did not cause permanent damage to the photosynthetic system, and the decrease in the photosynthetic activity as a result of the pH drop can be recovered when the pH is corrected.

**Keywords:** biogas upgrading; microalgae; carbon sequestration; pH

## 1. Introduction

Today, producing biogas from different wastes is a highly recommended strategy contributing to the production of non-conventional renewable energy, so that organic matter contained in effluents and wastes can be used as substrates in anaerobic reactors. In this sense, produced biogas can be injected into the natural gas grid, used as vehicular fuel, or burned to obtain thermal/electrical energy. Depending on final use, purification of biogas can be required, so a biogas upgrading system must be coupled to a biogas reactor to assuring the high methane content and removal of carbon dioxide ($CO_2$), hydrogen sulfide ($H_2S$), and other gases (nitrogen ($N_2$), ammonium ($NH_3$), siloxane, etc.) [1].

A biological technology for upgrading biogas is microalgae culture, where due to photosynthetic activity, these microorganisms can uptake $CO_2$ from biogas as a carbon

source. A special configuration for biogas upgrade systems through microalgae culture is the two-stage system, where an absorption unit is coupled with microalgae culture. In this configuration, the contact between the biogas and microalgae takes place in the absorption unit. Thus, the operation of an open photobioreactor connected to a counterflow bubble column (absorption unit) for carbon dioxide ($CO_2$) absorption represents a feasible alternative for biogas upgrade. The result of this two-stage system is an upgraded biogas with low $CO_2$ and oxygen ($O_2$) levels. Figure 1 shows a schematic representation of such a process.

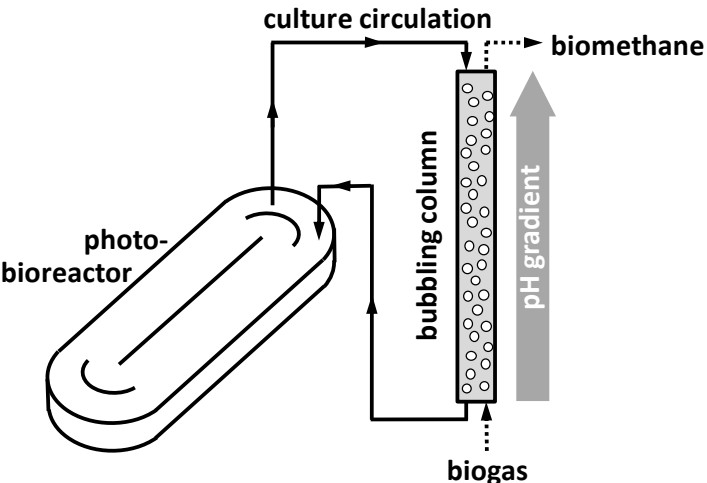

**Figure 1.** Schematic representation of the biogas upgrading process, involving microalgae culture and an absorption column.

To date, there are no operating full-scale installations for algal biogas upgrading, and researchers have focused their work on enhancing the performance of algal biogas upgrading systems in indoor/outdoor conditions at lab/pilot scale, modifying the operational parameters of the absorption unit (L/G) ratio [2], decreasing the $O_2$ concentration in biogas treated through the nitrification bacteria process in an algal photobioreactor [3], the addition of a trickling filter for bacterial oxygen removal [4], biogas diffusers [5], simultaneous $CO_2$ and $H_2S$ removal [6], biogas supply regime [7], etc. All this research indicates efforts carried out to achieve a successful implementation of biogas upgrading through microalgae at full scale.

However, although it has been widely reported that the utilization of a counterflow column improves the quality of upgraded biogas, the microalgal culture is exposed to important pH changes when circulating through the column [8]. Such pH changes are higher when reducing the flow, and pH variations are likely to produce a metabolic change in the microalgae culture. In this sense, the pH is an important parameter in the operation of a photosynthetic biogas upgrading system because the pH influences the inorganic carbon equilibrium [9] and the microalgae activity [10]. When $CO_2$ is dissolved in the aqueous phase, its inorganic carbon species depends on the pH [11–13]. The inorganic carbon dissociation causes the release of $H^+$, and as a result, pH decreases. The pH reduction affects the microalgae activity because most microalgae culture grows at a pH range of 7–9, with an optimal pH between 8.2 and 8.7 [14]. Most of the carbon is in bicarbonate form in a pH range of 7–9. Although $CO_2$ is the substrate of the Rubisco enzyme, microalgae cells can use bicarbonate as a carbon source [15]. Bicarbonate can be transformed into dissolved $CO_2$ by the enzyme carbonic anhydrase (CA) [16–18]. Although $CO_2$ dissolution causes a pH decrease, the activity of CA causes pH increases outside the cell due to the transport of hydroxide ions outside the cell in association with the capture of $H^+$ ions for the interior of the thylakoid membranes [19]. In this sense, research for biogas upgrading through microalgae culture had shown a pH drop in the absorption unit, where the pH has decreased up to 2 points as a result of $CO_2$ dissolution in the liquid phase [3,20]. On

the contrary, as already commented, it is expected that a pH increase takes place in the photobioreactor because of the photosynthetic activity.

Given the importance of the pH on microalgae cultivation, supplementary research is needed to clarify the potential effect of these dynamic pH conditions on the culture. Thus, the aim of this work is to evaluate the effect of the pH gradients expected in the column on the photosynthetic activity and PSII quantum yield. Photosynthetic activity refers to the oxygen released by the microalgae from water photolysis under saturating photosynthetically active radiations (PAR) [21]. The Photo-system II (PSII) quantum yield (Fv/Fm) reflects the performance of the photochemical processes in PSII. The PSII quantum yield ranges from 0.65 to 0.80 in healthy microalgae cultures [22]. Both analyses allow testing the condition of the photosynthetic system and the cell viability.

## 2. Materials and Methods

### 2.1. Microalgae Culture

The microalga *Chlorella sorokiniana* was obtained from the culture collection of Central Research Services (CIDERTA) of the University of Huelva, Huelva, Spain. Microalgae were cultivated using a modified M-8a medium [23]. All assays were carried out considering that the optimal pH of *C. sorokiniana* is 7.0, and they were grown until the stationary phase.

### 2.2. Experimental Procedure

Batch photobioreactors of 200 mL were used and inoculated from previous microalgae batch culture (Section 2.1). applying a light intensity of 90 µmol m$^{-2}$ s$^{-1}$. The effect of pH changes on the microalgae culture was evaluated through two experiments, which considered a shift in pH caused by the addition of HCl and $CO_2$ injection. Thus, these experiments simulated the expected pH changes evidenced when real biogas is bubbled in counterflow columns in microalgae culture:

- Change in the pH by addition of HCl 3.7%. Three conditions were evaluated: control culture at pH 7.0; pH change from 7.0 to 5.0 when the culture was exposed to light; pH change from 7.0 to 5.0 when the culture was in darkness. Moreover, for these three conditions, a biomass concentration of 0.5 and 1.3 g L$^{-1}$ was evaluated. The effect of the pH changes was evaluated through the photosynthetic activity and PSII quantum yield analysis.
- Change in the pH by $CO_2$ injection. Three conditions were applied: control culture at pH 7.0 (dissolved inorganic carbon concentration of 12 mM); culture exposed to $CO_2$ injection and light; culture exposed to $CO_2$ injection in darkness. $CO_2$ was bubbled into the microalgae culture lowering the pH value to pH 5.8. Then, the $CO_2$ injection was stopped and the pH, PSII quantum yield, and photosynthetic activity were determined. A biomass concentration of 1 g L$^{-1}$ was used. All assays were carried out in triplicate.

### 2.3. Determination of the Apparent Affinity of Microalgae

The use of inorganic carbon by *C. sorokiniana* was studied by photosynthetic activity (PA) kinetics (oxygen release) at pH 5.0 and pH 7.0, applying different dissolved inorganic carbon (DIC) concentrations into the electrode. The inorganic carbon was added in the form of $NaHCO_3$, partly converted into $CO_2$ as a function of the pH according to the chemical equilibrium $NaHCO_3/CO_2$ in water. The initial oxygen release rate was registered for each $NaHCO_3$ concentration added [21]. The apparent affinity constant (KDIC) for inorganic carbon was calculated from a graph of 1/PA versus 1/[DIC], according to Equation (1).

$$\frac{1}{PA} = \frac{K_{DIC}}{PA_{max}} \cdot \frac{1}{[DIC]} + \frac{1}{PA_{max}} \tag{1}$$

### 2.4. Analytical Methods

The photosynthetic activity was determined by oxygen evolution and standardized by chlorophyll content. Thus, photosynthetic activity was computed as µmol h$^{-1}$ µg$^{-1}$

chlorophyll. For oxygen determination, a Clark-type electrode was used. Oxygen release measurements were made under saturating white light (750 µmol m$^{-2}$ s$^{-1}$) or darkness (endogenous respiration) at 25 °C [24]. Chlorophyll content was determined by methanol extraction and visible spectrophotometry. The chlorophyll concentration in the extract was calculated by modifying Arnon's Equation (2) [25]:

$$Total\ Chlorophyll\ (\mu g/mL) = 20.2(A_{645nm}) + 8.02(A_{663}) \tag{2}$$

The PSII maximum quantum yield was measured using pulse amplitude modulation (PAM) fluorometry with the saturating-pulse technique [26]. The DIC was analyzed by alkalinity determination according to the method 4500 of standard methods [27].

*2.5. Statistical Analysis*

The photosynthetic activity and PSII quantum yield for different tested conditions were analyzed through independent sample t-student tests. All these analyses were carried out using the statistical software SPSS19. A significance level of 5% ($\alpha = 0.05$) and N = 3 were used in all cases.

## 3. Results

Figure 2 shows the photosynthetic activity and PSII quantum yield of the microalgae culture when the pH decreased from 7.0 to 5.0 after the addition of HCl, at two biomass concentrations of 0.5 and 1.3 g L$^{-1}$. The photosynthetic activity of the light-exposed culture decreased after 100 min for both biomass concentrations when the pH changed from 7.0 to 5.0 (Figure 2A,B). On the other hand, although the PSII quantum yield slightly decreased for both biomass concentrations, it remained around 0.6 and 0.7 (Figure 2C,D), such values correspond to healthy batch microalgae cultures [22]. When the pH was adjusted from pH 7.0 to pH 5.0 at 0.5 g L$^{-1}$ (Figure 2A), the photosynthetic activity of the culture recovered, demonstrating that the cells did not suffer permanent damage. On the other hand, when the pH was changed in the conditions of darkness, no significant changes in the photosynthetic activity were observed.

Figure 3 shows the photosynthetic activity and PSII quantum yield of the microalgae culture when the pH was decreased from 7.0 to 5.8 by CO$_2$ injection, simulating the process occurring in the column (the DIC concentration in the microalgae culture was tripled because of CO$_2$ bubbling). In contrast to the situation of the pH change by only acid addition (Figure 2A,B), when the pH was reduced by CO$_2$ injection, the photosynthetic activity decreased immediately in the culture exposed to light and darkness (Figure 3A). This result could mean that CO$_2$ inhibits the microalgae activity by a different mechanism for pH decrease. The photosynthetic activity was recovered after CO$_2$ injection. However, once CO$_2$ injection was stopped, the pH increased and the DIC concentration decreased in the culture medium due to both carbon fixation and desorption processes. In this sense, similar results were obtained with *Chlorella sp.*, whose growth was inhibited when it was exposed to high CO$_2$ concentrations, but the growth reappeared when the concentration was decreased [28].

As shown in Figure 3B, the CO$_2$ injection caused an increase in the PSII quantum yield. An increase in PSII yield means that a higher percentage of the absorbed light energy was used in the photochemical process. This response could be attributed to an increase in the demand for reducing power (NADPH) to fix and reduce the higher carbon inorganic concentration in the culture medium [29].

To study the preferred inorganic carbon source of *C. sorokiniana*, the apparent affinity constant for the dissolved inorganic carbon was determined. Figure 4 shows the photosynthetic activity as a function of the inorganic carbon concentration provided in the electrode cube at pH 7.0 and pH 5.0. The apparent affinity constant for dissolved inorganic carbon (KDIC) was 0.9 µM at pH 5 and 112.0 µM at pH 7. The NaHCO$_3$ added into the algal samples at pH 5 was mostly in the form of CO$_2$. Therefore, the lower apparent KDIC value at pH 5 than pH 7 suggests that this microalga had a higher affinity for CO$_2$ than HCO$_3^-$.

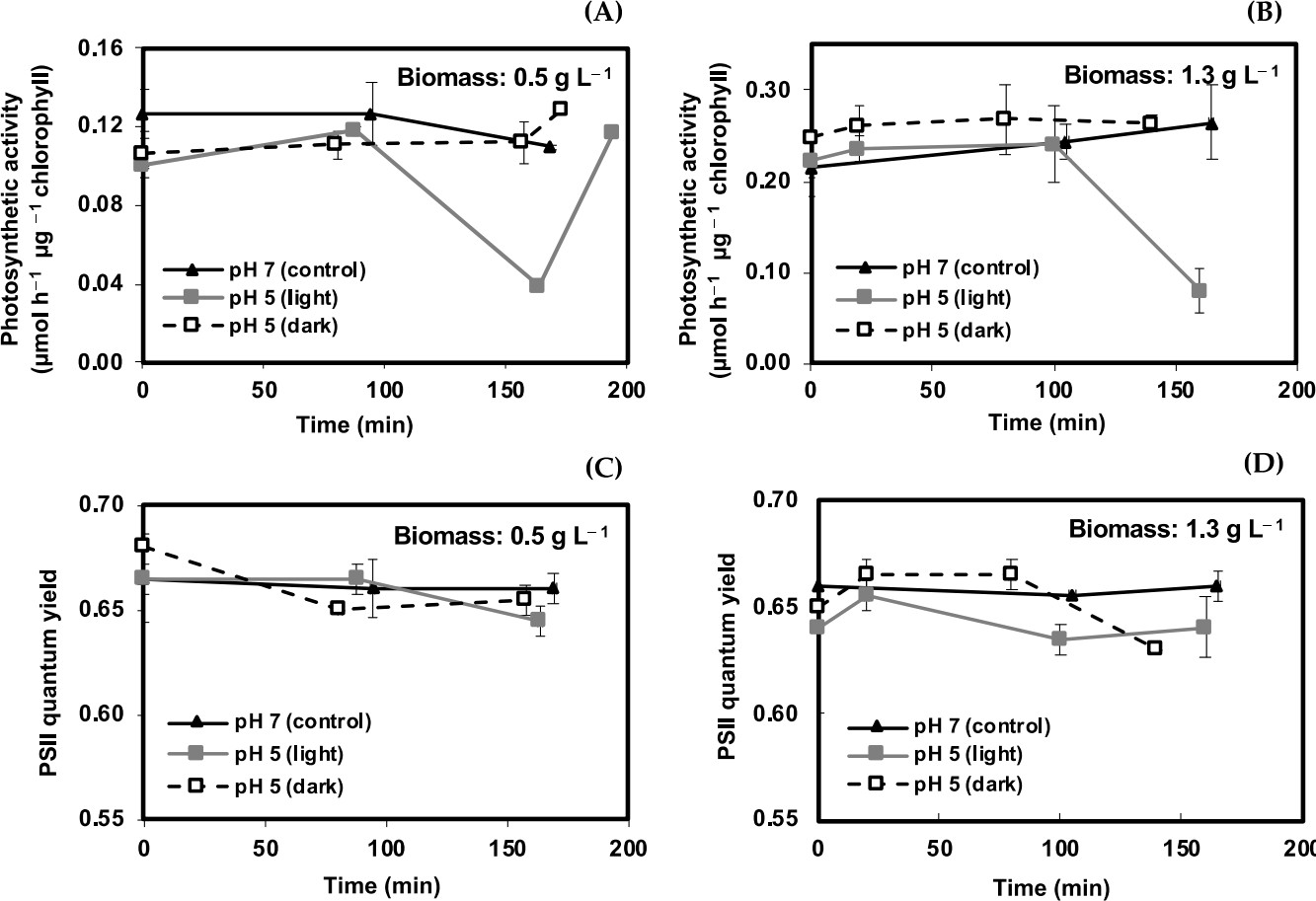

**Figure 2.** Effect of the pH change on the microalgae activity by HCl addition. (**A**) Photosynthetic activity when a biomass concentration of 0.5 g L$^{-1}$ was used. (**B**) Photosynthetic activity when a biomass concentration of 1.3 g L$^{-1}$ was used. (**C**) PSII quantum yield when a biomass concentration of 0.5 g L$^{-1}$ was used. (**D**) PSII quantum yield when a biomass concentration of 1.3 g L$^{-1}$ was used.

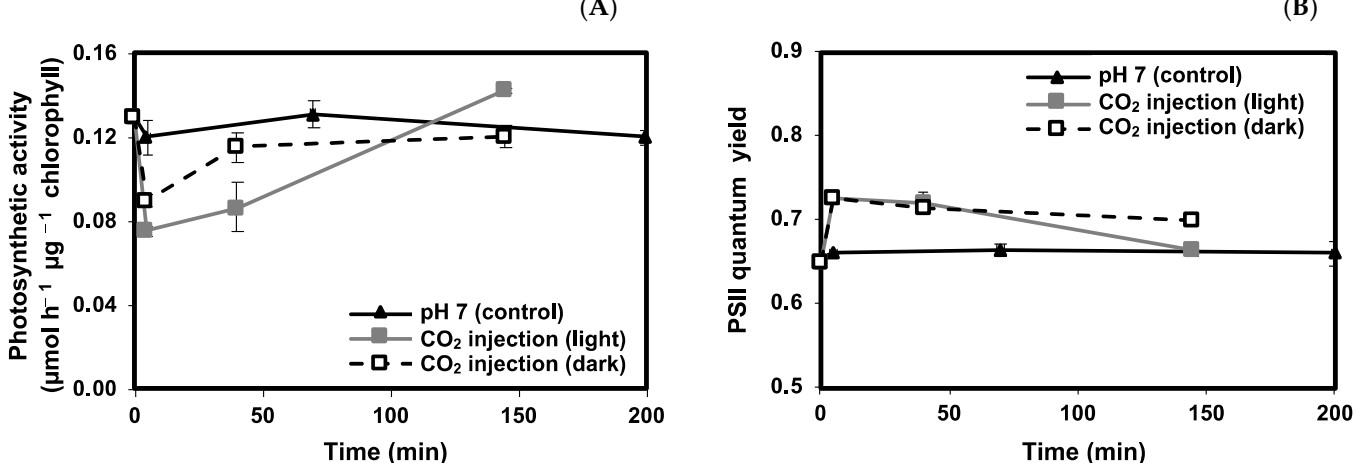

**Figure 3.** Effect of the pH and DIC concentration changes on microalgae activity. (**A**) Photosynthetic activity when $CO_2$ injection was applied. (**B**) PSII quantum yield when $CO_2$ injection was applied.

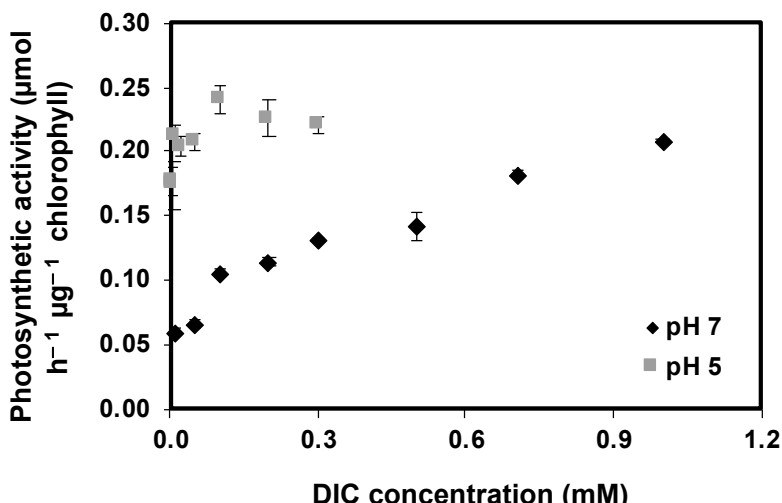

**Figure 4.** Photosynthetic activity curves of *Chlorella sorokiniana* at pH 7.0 and pH 5.0, as a function of the DIC concentration.

## 4. Discussion

According to the results in Figure 2A, in a counter-flow column for biogas upgrading, a pH shift takes place from 7.0 to 5.0; in light conditions, it is expected that this condition affects photosynthetic activity after 100 min. However, the counter-flow column is operated considering a residence time less than 20 min, which indicates that the decrease in photosynthetic activity by a pH shift will not take place [30]. Moreover, the absorption columns for biogas upgrading through microalgae are able to operate in dark without increasing the oxygen concentration in upgraded biogas [4,30]

In relation to the results shown in Figure 3, it can be supposed that if a pH drop takes place, such as has been evidenced when biogas is injected into the microalgae culture in the column, the photosynthetic activity of algal cells could decrease. However, in this case, the cells do not suffer damage in their photosynthetic system, maintaining a high value of PSII quantum yield, such as the result obtained in this research. When microalgae return to the photobioreactor, the DIC concentration decreases due to photosynthesis and desorption, and the microalgal cells recover their photosynthetic activity.

The result obtained, as shown in Figure 3, indicates that $CO_2$ would be the inorganic carbon source preferred by *C. sorokiniana*. This result agrees with Williams and Colman [31], who indicated that *C. saccharophila* had an affinity for $CO_2$, which was 160 times greater than that for $HCO_3^-$. The highest affinity at pH 5 could suggest the expression of some concentrating mechanisms of $CO_2$ that could facilitate its fixation by Rubisco. According to Tsuzuki, Shiraiwa, and Miyachi [32], there are two possible ways by which $CO_2$ may be supplied to the Chlorella surface: $CO_2$ can be supplied from the culture medium by simple diffusion (direct supply of $CO_2$) or $HCO_3^-$ formed from $CO_2$ can be converted again into $CO_2$ via the enzyme carbonic anhydrase (CA) and incorporated by the algal cells (indirect supply of $CO_2$).

Additional research must be carried out to clarify the mechanisms affected in the algal cells due to the injection of a gas with a high $CO_2$ concentration and/or a decrease in the pH, considering that $CO_2$ is the preferred source of carbon for Chlorella. A possible substrate inhibition may occur in an enzyme involved in the mechanism of carbon consumption. On the other hand, because of the circulating flow between the photobioreactor and the column, microalgae cells can be exposed to pH and DIC gradients several times during the operation of the system. Thus, it would be interesting to study whether these repeated changes had some additional effect on microalgae activity.

## 5. Conclusions

Based on the results of this study, it is expected that the pH gradients in an absorption column would not cause damage to the photosynthetic system of microalgae, because a high value of PSII efficiency remained, and the photosynthetic activity could be recovered. This means that for reported pH drops in the absorption column, photosynthetic activity will not be affected, which assures an efficient $CO_2$ uptake process and growth in microalgae culture.

$CO_2$ is the preferable source of carbon for *C. sorokinana*. However, additional research must be carried out to study the mechanisms that are affected in the algal cells when a gas with high $CO_2$ concentration is applied.

**Author Contributions:** Conceptualization, L.M., D.J. and C.V.; methodology, L.M., C.V. and M.C.; formal analysis, L.M. and M.C.; resources, D.J.; writing—original draft preparation, L.M.; writing—review and editing, M.C. and Á.T.-A.; supervision, D.J. and C.V. All authors have read and agreed to the published version of the manuscript.

**Funding:** This research was funded by FONDECYT-ANID CHILE, grant number 1120488, CRHIAM Centre (CONICYT/FONDAP) grant number 15130015, and VRIEA-PUCV grant number 039.315/2022.

**Institutional Review Board Statement:** Not applicable.

**Informed Consent Statement:** Not applicable.

**Data Availability Statement:** Not applicable.

**Acknowledgments:** The authors want to thank the FONDECYT project 1120488 (ANID, Chile), CRHIAM Centre (CONICYT/FONDAP/15130015), and VRIEA-PUCV 039.315/2022 for the financial support.

**Conflicts of Interest:** The authors declare no conflict of interest.

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
