# Peer review of "Effect of pH Change on the Microalgae-Based Biogas Upgrading Process"

_applsci, doi:10.3390/app122312194_

Round 1

Reviewer 1 Report

The document presented by the authors is an exciting approach to the pressing matter of using CO2 from biogas production. However, the authors failed to explain their selection of experiments since they do not use real biogas.

Also, the authors must consider extensive editing on the results since the exposed methodology claims to measure metabolites such as carotenoids, but their results are not presented. The same applies to DIC data. 

Abstract

Lines 3-9 are the same as lines 33-35 in the introduction. Please upgrade the abstract.

Methods

Line 83, please review the upper case writing of umol units.

Since the authors did not use real biogas in the experiments, in section 2.2, the authors must explain to the readers that the investigations will simulate the process that might happen in a bubbling column. 

Authors must clearly state why it is crucial to test the pH changes with different biomass concentrations in the bubbling column.

Results

In line 152, specify which strain of chlorella it was an “sp” or a known species.

The second time Chlorella sorokiniana appears in the document, it should be written in their condensed version, “C. sorokiniana.”

The authors state in line 114 that they measure chlorophylls and carotenoids. However, there are no data on carotenoid concentration or changes through the experimentation process.

The same occurs with the dissolved organic carbon; why did the authors add these methods if there is no relevant data?

Author Response

Response to Reviewer 1 Comments

Point 1: The document presented by the authors is an exciting approach to the pressing matter of using CO2 from biogas production. However, the authors failed to explain their selection of experiments since they do not use real biogas. Also, the authors must consider extensive editing on the results since the exposed methodology claims to measure metabolites such as carotenoids, but their results are not presented. The same applies to DIC data

Response 1: Such as mentioned by reviewer 1, no real biogas was used, and counter-flow column was not operated. This choice was founded on the fact that in another published article, the first author (Meier, L), reported that a pH gradient up to 2 was observed when microalgae were circulated from the counter-flow column to the photobioreactor. Based on this, this short communication was focused on responding if photosynthetic activity could be affected when pH changes take place in the counter-flow column coupled to the photobioreactor. Thus, this fact was clarified in this article, specifically in the introduction, and the discussion considered that results obtained under pH changes provoked by HCl addition and CO2 injection are representative of counter-flow column coupled to photobioreactor.

In relation to carotenoids, it was not measured, so that its methodology was deleted from the article. On the other way, DIC concentration was measured and DIC data was showed in Figure 4, where photosynthetic activity at pH 5.0 and 7.0 was evaluated considering different DIC concentrations (mM).  

Point 2: Lines 3-9 are the same as lines 33-35 in the introduction. Please upgrade the abstract.

Response 2: The indicated lines (3-9) were modified. New lines added are the following: “This technology provides up-graded biogas with high quality. In this sense, the microalgal uptake of CO2 from biogas in counterflow columns generates pH changes in microalgae culture.” Lines 33-35 were maintained.

Point 3: Line 83, please review the upper-case writing of umol units.

Response 3: units for light intensity correspond to µmol m-2 s-1. Thus, units were modified adding upper-case for meters and seg. New format for unit is as following: µmol m-2 s-1.

Point 4: Since the authors did not use real biogas in the experiments, in section 2.2, the authors must explain to the readers that the investigations will simulate the process that might happen in a bubbling column.

Response 4: For clarifying the carried-out assays where no real biogas was used for causing pH changes, the following phrase was added in section 2.2: “The effect of pH changes on the microalgae culture was evaluated through two experiments, which consider a shift in pH caused by the addition of HCl and CO2 injection. Thus, these experiments will simulate the expected pH changes evidenced when real biogas is bubbled in counterflow columns in microalgae culture:”

Point 5: Authors must clearly state why it is crucial to test the pH changes with different biomass concentrations in the bubbling column.

Response 5: As already mentioned, a counter-flow column was not operated, and this research was focused on changes at the photosynthetic level when pH shifted from 7.0 to 5.0 (pH changes were carried-out by HCl and CO2 addition in culture). In a previous article published by the author of this short communication (Meier, L.), pH of 2 point was evidenced the photobioreactor coupled with counter-flow column was operated for biogas upgrading. 

Point 6: In line 152, specify which strain of chlorella it was an “sp” or a known species.

Response 6: The microalgae used in research (18) was Chlorella Sp, hence “Sp” was added to Chlorella. Moreover, italic was added for microalgae name.

Point 7: The second time Chlorella sorokiniana appears in the document, it should be written in their condensed version, “C. sorokiniana.”

Response 7: After the first time that the microalgae Chlorella sorokiniana was named in the article, it name was changed by “C. sorokiniana” every time that it was named.

Point 8: The authors state in line 114 that they measure chlorophylls and carotenoids. However, there are no data on carotenoid concentration or changes through the experimentation process. The same occurs with the dissolved organic carbon; why did the authors add these methods if there is no relevant data?

Response 8: Carotenoid concentration was not measured, chlorophyll concentration was measured and used for calculating photosynthetic activity expressed as “µmol h-1 µg-1 chlorophyll”. This explanation was added to the methodology as follows (section 2.4): “The photosynthetic activity was determined by oxygen evolution and standardized by chlorophyll content. Thus, photosynthetic activity was computed as µmol h-1 µg-1 chlorophyll. For oxygen determination, a Clark-type electrode was used”. Photosynthetic activity values can be observed in Figures 2 – 4 (ordinate-axis). In relation to DIC concentrations, they were indicated in Figure 4. (abscissa-axis)

Reviewer 2 Report

Applsci-1920050 investigates the effect of pH change on the photosynthetic activity and PSII quantum yield in the counterflow column used for biogas upgrading. In addition, the authors also studied the effect of biogas flow rate, CO2 injection, and light/darkness on photosynthetic activity and PSII quantum yield.

Although the topic is interesting and has both academic and real-field importance, the following issues convinced me to reject this article directly.

 General comment:

The abstract should clearly state the research gap.

Please define all abbreviations at their first appearance in the text.

I'd like to see your key numerical findings at the end of the abstract.

A paragraph needs to add at the end of the introduction section and cover the following topic:

a- Limitation/drawback of our knowledge in this field

b- your suggested scenario to improve the knowledge

c- Novelty of the study

The conclusions section needs to rewrite.

The main parts of your references are not up-to-date. You only used two references published after 2017, while 25% of them were published before 2000.

Scientific concerns:

You reviewed no literature in the introduction section. It is necessary to review the related literature about the main topic of your study. The up-to-date references (published in the last 2017-2022) are preferred (like this one: https://doi.org/10.1016/j.eti.2022.102770).

It is not a good idea to only report the results and not analyze them. For instance:

a- You should provide some scientific reasons for the variation of activity and yield by biomass flow rate, pH, time, light, and darkness (Figures 2 and 3).

b- What do sharp descending/ascending, minimum/maximum, and sudden trend changes mean? 

c- What is the best condition for the biogas upgrading in your study?

Generally, it seems you did not have enough time to appropriately organize your research and analyze the results.

Reviewer 3 Report

The authors performed a study in which they cultivated microalgae in a photobioreactor that is connected to a counterflow bubble column for CO2 absorption, and evaluated the effect of pH gradients in the column after a) addition of HCl, and b) CO2 injection. I found this study to be interesting and generally well written, but I have some comments and questions. 

1. In the introduction the studies related to the given topic should be added - to give a short overview of the other related/similar studies. 

2. Subsection 2.2. Effect of pH change on the microalgae culture seems a bit inappropriate. Maybe the caption Experimental procedure would look more precise. 

3. There is no information about the duration (hours, days...) of the experiment - it is not quite clear does the microalgae continuously circulate through the column; do they pass through the column several times?  

Information about the temperature (in the photobioreactor and the column) that was maintained during the experiment is also missing. Regarding, the light/dark cycles of the microalgae - it is not clear how long these periods last.   

4. Please correct the name of the microalgae to italic and unify the pH values (e.g. 7 or 7.0) throughout the whole manuscript. 

5. Line 37: replace on with in

6. Line 51: remove on ("...on the carbon inorganic...")

7.  Line 83: correct unit (m-2 s-1)

8. Line 112: add space between 25 and degrees Celsius 

9. Line 122: add the after by, before addition (by the addition) 

10. Line 124: replace change with 

11. Line 129: replace darkness condition with dark conditions

12. Line 131: replace with with to (according to the results...)

13. Line 157: replace to photochemical process with in the photochemical process

14. Line 158: replace of with for (for reducing power)

15. Line 170: replace come back with return

Author Response

Response to Reviewer 3 Comments

Point 1: In the introduction, the studies related to the given topic should be added - to give a short overview of the other related/similar studies.

Response 1: Some references about related/similar studies were added in article (introduction section), according to the reviewer`s comment.

Point 2: Subsection 2.2. Effect of pH change on the microalgae culture seems a bit inappropriate. Maybe the caption Experimental procedure would look more precise.

Response 2: The title of Subsection 2.2 was modified according to the reviewer's comment. Thus, the new title of subsection 2.2 is “Experimental procedure”.

Point 3: There is no information about the duration (hours, days...) of the experiment - it is not quite clear does the microalgae continuously circulate through the column; do they pass through the column several times? 

Response 3: It is worthy to clarify that experiments carried-out in this research were focused on the effect of pH changes in microalgae culture, so that in other article published for first author (Meier, L.) , these pH changes in counter-flow column was measured, indicating that pH gradient up to 2 was obtained when microalgae circulated from column to photobioreactor (https://doi.org/10.1016/j.biombioe.2014.10.032). Thus, this short communication was focused on the evaluation of photosynthetic activity behavior when pH changes in microalgae culture takes place. For this purpose, pH changes were carried-out through the addition of HCl and direct CO2 injection. Thus, in these experiments, no real biogas was injected to the microalgae culture and the absorption column was not operated. In this sense, as other reviewer noticed and commented this fact, in this article was highlighted that “the absorption column was not operated, and no real biogas was injected for provoking pH changes” focusing only on the effect of pH changes on the photosynthetic activity.

Point 4: Information about the temperature (in the photobioreactor and the column) that was maintained during the experiment is also missing. Regarding, the light/dark cycles of the microalgae - it is not clear how long these periods last.

Response 4: Please, consider Response 3, which will clarify point 4.

Point 5: Please correct the name of the microalgae to italic and unify the pH values (e.g. 7 or 7.0) throughout the whole manuscript.

Response 5: The name of microalgae was modified to italic. Moreover, pH values were unified according to the reviewer's comment.

Point 6: Line 37: replace on with in

Response 6: Correction was made according to the reviewer`s comment.

Point 7: Line 51: remove on ("...on the carbon inorganic...")

Response 7: Correction was made according to the reviewer`s comment.

Point 8: Line 83: correct unit (m-2 s-1)

Response 8: unit was corrected according to the reviewer`s comment.

Point 9: Line 112: add space between 25 and degrees Celsius

Response 9: Modification was made according to the reviewer`s comment.

Point 10: Line 122: add the after by, before addition (by the addition)

Response 10: Modification was made according to the reviewer`s comment.

Point 11: Line 124: replace change with changed

Response 11: Modification was made according to the reviewer's comment.

Point 12: Line 129: replace darkness condition with dark conditions

Response 12: Modification was made according to the reviewer's comment.

Point 13: Line 131: replace with with to (according to the results...)

Response 13: Modification was made according to the reviewer's comment.

Point 14: Line 157: replace to photochemical process with in the photochemical process

Response 14: Modification was made according to the reviewer's comment.

Point 15: Line 158: replace of with for (for reducing power)

Response 15: Modification was made according to the reviewer's comment.

Point 16: Line 170: replace come back with return

Response 16: Modification was made according to the reviewer's comment.

Reviewer 4 Report

This is an interesting study and You have collected a unique dataset using valid experimental methodology. The paper is generally well written. However, in my opinion it has some shortcomings in its structure especially in regard to the way that datasets are presented.  Particularly, parts of the datasets are presented and discussed in Results section while other data is presented in the Discussion section. In my opinion structurally, all experimental datasets need to be presented in the Result section while discussion and transfer of technology should be in the Discussion section. Therefore, I suggest some revisions are necessary. Also, English language and style require proofreading.

You may see my comments in the attached copy of the manuscript.

Kind regards,

Assoc. prof. Sanja Tomsic

Author Response

Response to Reviewer 4 Comments

Point 1: Particularly, parts of the datasets are presented and discussed in Results section while other data is presented in the Discussion section. In my opinion structurally, all experimental datasets need to be presented in the Result section while discussion and transfer of technology should be in the Discussion section.

Response 1: Effectively, results indicated in Figure 3 are located in the Discussion section and discussion paragraphs are located in Result section. Thus,  according to the reviewer comment, these paragraph were located in the correct section.

Point 2:  Therefore, I suggest some revisions are necessary. Also, English language and style require proofreading.You may see my comments in the attached copy of the manuscript.

Response 2: Multiple corrections were carried out considering the reviewer's comment attached to the manuscript.

Reviewer 5 Report

The manuscript entitled “Effect of pH change on microalgae-based biogas upgrading process” (Manuscript ID: applsci-1920050) aims to know the effect of pH change on the photosynthetic activity and PSII quantum yield in Chlorella sorokiniana. The study design and the methodologic approaches used are appropriate. However, the manuscript requires some improvement in the results and discussion sections based on updated scientific information to be considered for publication in the Journal of Applied Sciences.

Some suggestions/recommendations are the following:

ABSTRACT

1.     Page 1, line 26, could change the word “moving” by “decreasing”

2.     Page 1, line 27, the sentence “Moreover, light and darkness…..pH drop” could be rewritten to clarify the point.

3.     The authors could verify that the keywords are correct. Please, check if the keywords are accurate using this link: https://meshb.nlm.nih.gov/ or identify keywords in the abstract section using the MeSH® on-demand online tool (https://meshb.nlm.nih.gov/MeSHonDemand).

INTRODUCTION

4.     Page 2, lines 66-74, the Authors could rewrite this paragraph; the sentences are significantly longer. Also, the last sentence, “All these articles…..at full-scale,” is unclear.

5.     Page 3, lines 86-107, change the expression “carbon inorganic” to “inorganic carbon”.

6.     Page 3, line 93, change “it can dissociate into bicarbonate…” to “it is converted into bicarbonate…”. Because when CO2 reacts with the water molecule is formed carbonic acid (H2CO3). Further, depending on the medium pH, carbonic acid is dissociated into HCO3- and H+. Also, HCO3- could be dissociated into CO32- + H+.

7.     Page 3, lines 110-111, the Authors indicate that “no research indicating the effect of pH…has been reported”. However, in the scientific literature exist, several publications about this topic. Authors could revise some of the following publications:

1) https://link.springer.com/article/10.1007/s00253-013-5035-2

2) https://www.mdpi.com/2227-9717/9/5/820

3) https://link.springer.com/article/10.1007/BF02949280

            And rewrite the sentence.

MATERIALS AND METHODS

8.     The subtitles are confusing; these could be changed to more appropriate ones.

9.     Provide a citation for calculations of the apparent affinity constant (KDIC) for inorganic carbon.

10.  Authors could indicate the growth phase (lag, exponential, etc) of the microalgae culture chosen to make the assays.

11.  The authors could provide the equations used to determine chlorophyll content.

12.  The Authors could include the determination of inorganic carbon (i.e., CO2, HCO3- and CO32-) concentrations through the assays. In addition, the Authors could include measurements of oxygen concentrations.  

13.  The authors could use biological and technical replicates for the assays. The authors could clarify this point.

14.  Authors could include statistical methods to determine the normality of a data set (i.e., Kolmogorov-Smirnov test, Shapiro-Wilk test, etc.) and determine if exist differences statistically significant among treatments tested.

RESULTS AND DISCUSSION

15.  In general, the writing and paragraph structure should be improved

16.  The resolutions and qualities of the figures need to be significantly enhanced; its current version is blurry.

Author Response

Response to Reviewer 5 Comments

Point 1: Page 1, line 26, could change the word “moving” by “decreasing”

Response 1: Change was carried out according to the reviewer comment.

Point 2: Page 1, line 27, the sentence “Moreover, light and darkness…..pH drop” could be rewritten to clarify the point.

Response 2: This phrase was modified according to reviewer comment.

Point 3: The authors could verify that the keywords are correct. Please, check if the keywords are accurate using this link: https://meshb.nlm.nih.gov/ or identify keywords in the abstract section using the MeSH® on-demand online tool (https://meshb.nlm.nih.gov/MeSHonDemand).

Response 3: keyword were checked and keyword "CO2 removal" was changed by "Carbon sequestration"

Point 4: Page 2, lines 66-74, the Authors could rewrite this paragraph; the sentences are significantly longer. Also, the last sentence, “All these articles…..at full-scale,” is unclear.

Response 4: this paragraph was rewrite. "Full-scale" is a typical concept applied to reactor scale-up, so that this phrase was not modified, so that authors think that it is well understood by other scientific/engineers.

Point 5:Page 3, lines 86-107, change the expression “carbon inorganic” to “inorganic carbon”.

Response 5: expresión "carbon inorganic" was modified to "inorganic carbon" according to the reviewer comment.

Point 6: Page 3, line 93, change “it can dissociate into bicarbonate…” to “it is converted into bicarbonate…”. Because when CO2 reacts with the water molecule is formed carbonic acid (H2CO3). Further, depending on the medium pH, carbonic acid is dissociated into HCO3- and H+. Also, HCO3- could be dissociated into CO32- + H+.

Response 6: review comment is right, so phrase was modified according to the reviewer comment

Point 7: Page 3, lines 110-111, the Authors indicate that “no research indicating the effect of pH…has been reported”. However, in the scientific literature exist, several publications about this topic. Authors could revise some of the following publications:

1) https://link.springer.com/article/10.1007/s00253-013-5035-2

2) https://www.mdpi.com/2227-9717/9/5/820

3) https://link.springer.com/article/10.1007/BF02949280

            And rewrite the sentence.

Response 7: This phrase was deleted from manuscript.

Point 8: The subtitles are confusing; these could be changed to more appropriate ones.

Response 8: some subtitles were changed and other were maintained.

Point 9:Provide a citation for calculations of the apparent affinity constant (KDIC) for inorganic carbon.

Response 9: citation [21] is part of the sentence indicating apparent affinity constant.

Point 10: Authors could indicate the growth phase (lag, exponential, etc) of the microalgae culture chosen to make the assays.

Response 10: growth phase was indicated in section according to the reviewer comment.

Point 11: The authors could provide the equations used to determine chlorophyll content.

Response 11: Chlorophyll equation was added according to the reviewer comment.

Point 12: The Authors could include the determination of inorganic carbon (i.e., CO2, HCO3- and CO32-) concentrations through the assays. In addition, the Authors could include measurements of oxygen concentrations.

Response 12: Authors thank the reviewer comment but we think that it is not necessary to include oxygen concentration measurements.

Point 13: The authors could use biological and technical replicates for the assays. The authors could clarify this point.

Response 13: authors added a sentence indicating that all assays were carried out in triplicates.

Point 14: Authors could include statistical methods to determine the normality of a data set (i.e., Kolmogorov-Smirnov test, Shapiro-Wilk test, etc.) and determine if exist differences statistically significant among treatments tested.

Response 14: the required information was added in section "2.4 Statistical analysis".

Point 15: In general, the writing and paragraph structure should be improved

Response 15: Structure paragraph was modified in some cases for clarify the information and some gramatical mistake were corrected.

Point 16:The resolutions and qualities of the figures need to be significantly enhanced; its current version is blurry

Response 16: authors modified actual Figures, adding high resolution images according to the reviews comment.

Round 2

Reviewer 1 Report

The authors did improve the document, which is now easier to understand.

Also, all the suggestions, on the improvement of the method section were done, which gives a clear view of the problem to be solved by the paper

Author Response

The authors of this article thank reviewer 1 for the new comments. As no new suggestion has been indicated, this article was not modified.

Reviewer 2 Report

Reject.

Author Response

As new comments/suggestions have not been indicated by reviewer 2, no article changes can be made. The authors thank reviewer 2 for spending time reviewing this article.

Reviewer 5 Report

The new version of the manuscript has been improved significantly